# New Frontiers in Autoimmune Diagnostics: A Systematic Review on Saliva Testing

**DOI:** 10.3390/ijerph20105782

**Published:** 2023-05-10

**Authors:** Silvia Grazietta Foddai, Massimo Radin, Alice Barinotti, Irene Cecchi, Elena Rubini, Marta Arbrile, Ester Mantello, Elisa Menegatti, Dario Roccatello, Savino Sciascia

**Affiliations:** 1University Center of Excellence on Nephrologic, Rheumatologic and Rare Diseases (ERK-Net, ERN-Reconnect and RITA-ERN Member) with Nephrology and Dialysis Unit and Center of Immuno-Rheumatology and Rare Diseases (CMID), Coordinating Center of the Interregional Network for Rare Diseases of Piedmont and Aosta Valley, Department of Clinical and Biological Sciences, University of Turin, 10154 Turin, Italy; silviagrazietta.foddai@unito.it (S.G.F.); massimo.radin@unito.it (M.R.); alice.barinotti@unito.it (A.B.); irene.cecchi@unito.it (I.C.);; 2Department of Clinical and Biological Sciences, School of Specialization of Clinical Pathology, University of Turin, 10124 Turin, Italy; marta.arbrile@unito.it

**Keywords:** saliva, autoimmune disorders, testing feasibility

## Abstract

(1) Background: Immunological laboratory testing is known to be complex, and it is usually performed in tertiary referral centers. Many criticalities affect diagnostic immunological testing, such as limited availability, the need for specifically trained laboratory staff, and potential difficulties in collecting blood samples, especially in the most vulnerable patients, i.e., the elderly and children. For this reason, the identification of a new feasible and reliable methodology for autoantibody detection is urgently needed. (2) Methods: We designed a systematic review to investigate the available literature on the utilization of saliva samples for immunological testing. (3) Results: A total of 170 articles were identified. Eighteen studies met the inclusion criteria, accounting for 1059 patients and 671 controls. The saliva collection method was mostly represented by passive drooling (11/18, 61%), and the most frequently described methodology for antibody detection was ELISA (12/18, 67%). The analysis included 392 patients with rheumatoid arthritis, 161 with systemic lupus erythematosus, 131 with type 1 diabetes mellitus, 116 with primary biliary cholangitis, 100 with pemphigus vulgaris, 50 with bullous pemphigoids, 49 with Sjogren syndrome, 39 with celiac disease, 10 with primary antiphospholipid syndromes, 8 with undifferentiated connective tissue disease, 2 with systemic sclerosis, and 1 with autoimmune thyroiditis. The majority of the reviewed studies involved adequate controls, and saliva testing allowed for a clear distinction of patients (10/12 studies, 83%). More than half of the papers showed a correlation between saliva and serum results (10/18, 55%) for autoantibody detection, with varying rates of correlation, sensitivity, and specificity. Interestingly, many papers showed a correlation between saliva antibody results and clinical manifestations. (4) Conclusions: Saliva testing might represent an appealing alternative to serum-based testing for autoantibody detection, considering the correspondence with serum testing results and the correlation with clinical manifestations. Nonetheless, standardization of sample collection processing, maintenance, and detection methodology has yet to be fully addressed.

## 1. Introduction

Using saliva samples instead of peripheral blood major advantages; for example, sample collection does not require trained staff, represents a low or minimally invasive procedure, and is generally well-received by patients, especially the pediatric and elderly populations. In addition, saliva samples can be easily stored and directly shipped from the homes of patients to central laboratories for analysis, limiting the need for patient mobility, with a positive impact on the quality of life both of patients and caregivers [1]. Interest in saliva-based diagnostic techniques has increased since the 1990s with the aim of improving accessibility to human immunodeficiency virus testing [2]. Over the last few years, the need for a feasible and reliable alternative to blood testing for antibodies and/or autoantibody detection attracted attention during the COVID-19 pandemic, when an affordable, user-friendly assay, such as saliva testing, was demanded to quickly identify affected subjects [3].

Saliva testing has demonstrated its applicability in different settings, for example, for detection of oral cancer such as squamous cell carcinoma [4], identification of infectious diseases (helicobacter pylori [5], hepatitis [6], papilloma virus [7], etc.), hormone monitoring [8,9,10], and screening for chronic kidney disorders [11]. Interesting results have emerged in myocardial infarction [12] and neurodegenerative disorder [13] detection through saliva specimen testing and in diabetes glucose level monitoring [14]. Furthermore, saliva testing has facilitated an increase in research on periodontal disorders [15] and caries prediction [16]. Moreover, saliva testing has been useful for drug level monitoring, such as in patients with epilepsy [17], and proved its suitability for forensic medicine and toxicology in testing of drugs of abuse [18].

Although saliva testing has a wide range of clinical applications, its use in clinical facilities is still limited by some intrinsic features; saliva composition (mostly water and small amounts of protein, electrolytes, urea, ammonia, glucose, free fatty acids, triglycerides, amino acids, white blood cells, epithelial cells, cytokines, nucleic acids, etc.) [19] precludes the feasibility of coagulation tests, as wells blood cell counts and blood gas level assessment. Additionally, the cutoff for evaluating analytes and compounds in saliva differs from that of serum. On the one hand, some analytes may naturally be more concentrated in saliva than in serum, leading to an overestimation of the substance concentration in the absence of an appropriate correction coefficient; on the other hand, some saliva tests may underestimate the true titer of the analyte under investigation [20], and interference-causing compounds (such as tea, coffee, food, etc.) can impact on the testing result [21]. Despite the mentioned drawbacks associated with the potential use of saliva testing in routine analysis, the simplicity of the methodology has attracted the interest of researchers and practitioners, and autoantibody testing represents a feasible area of implementation (no need for full blood for screening, necessity of reducing diagnostic delay, monitoring based on routine autoantibody testing, etc.). The aim of this systematic review was to investigate the current knowledge on autoantibody testing using saliva samples in the context of autoimmune disorders.

## 2. Materials and Methods

### 2.1. Systematic Review Process

A detailed literature search was developed *a priori* to identify articles that report findings on the utilization of saliva to detect autoantibodies for diagnostic purposes.

The following keywords and subject terms were included: (“saliva” [MeSH Terms] OR (“saliva” [All Fields]) AND (“autoantibodies” [MeSH Terms] OR (“autoantibodies” [All Fields]) AND (“autoimmunity” [MeSHTerms] OR (“autoimmunity” [All Fields])) AND “1967/01/01” [PDat]: “2022/08/25”[PDat]).]). The search strategy was applied to Ovid MEDLINE, In-Process, and other non-indexed citations for recent years. Data reported prior to the analysis was also retrieved by identifying papers from citation references.

The following inclusion criteria were applied to select studies for the review:(a)Original works written exclusively in English;(b)Testing in humans;(c)At least 10 patients enrolled and tested;(d)Detailed description of the techniques used for autoantibody detection.

Studies that met the inclusion criteria were systematically analyzed by two independent reviewers (S.G.F. and M.R.). Disagreements were resolved by consensus; if consensus could not be achieved, a third party (S.S.) provided an assessment of eligibility.

As the data on eligibility were dichotomous (eligible: yes/no), agreement at both the title and abstract review and the full-article review stages was determined by calculation of Cohen’s kappa coefficient (k > 0.8). Papers that did not meet the inclusion criteria and papers with a lack of data were considered ineligible and excluded from the analysis. The literature search strategy is shown in Figure 1. The presence of duplicates was assessed through EndNote 20 (Clarivate, London, UK) verification.

### 2.2. Data Extraction and Synthesis

Data from the eligible studies were independently extracted and listed by two reviewers (S.G.F. and M.R.) in Table 1 and Table 2. In the case of disagreement after discussion, a third party (S.S.) was consulted. Data extracted and summarized included study design, population type and numerosity, type and isotype of tested antibodies, methodology for antibody detection, saliva testing performance, saliva–serum correlation, and association of saliva testing results with disease clinical manifestations. The variability of included studies regarding the type of population analyzed, the type of antibodies tested, and the methodology applied did not allow for a pooled synthesis of the results. Therefore, all the results are discussed in a descriptive manner. The present study was performed according to PRISMA guidelines [22].

## 3. Results

A total of 170 articles were identified through the literature search and screened as detailed above. The search was conducted on 25 August 2022, and the time span of the 164 initially identified papers ranged from 1967 to 2022. A flow chart of the literature search strategy is reported in Figure 1. Eighteen studies met the inclusion criteria, accounting for 1059 patients and 671 controls (including 538 healthy controls (HCs), 42 oral lichen planus patients (OLP), and 91 first-degree patient relatives and were included in the final analysis [23,24,25,26,27,28,29,30,31,32,33,34,35,36,37,38,39,40]. The analysis included a wide range of autoimmune disorders, namely 161 cases of systemic lupus erythematosus (SLE), 10 cases of primary antiphospholipid syndrome (PAPS), 2 cases of systemic sclerosis (SSC), 1 case of autoimmune thyroiditis, 8 cases of undifferentiated connective tissue disorder (UCTD), 49 cases of Sjogren syndrome (SjS), 392 cases of rheumatoid arthritis (RA) (grouped under CTDs labelling), 100 cases of pemphigus vulgaris (PV), 50 cases of bullous pemphigoid (BP), 116 cases of primary biliary cholangitis (PBC), 131 cases of type 1 diabetes mellitus (T1DM), and 39 cases of celiac disease. Articles were divided into thematic categories: rheumatologic disorders (eight studies (44%): one on SLE, one on CTDs, two on sicca syndrome, and four on RA), dermatological disorders (three studies (17%): two on PV and one on BP), gastrointestinal and endocrinologic disorders (seven studies (39%): three on PBC, one on celiac disease, and three on T1DM).

### 3.1. Methodologies Applied

The vast majority of the studies (12 of 18 (67%)) used ELISA for their analysis [[23,27,28,29,30,31,32,33,35,37,38,40]. In detail, 10 of 12 employed commercial ELISA kits. Moreover, the following techniques were used in two studies each: radio immunoprecipitation assay (RIPA) [34,36], indirect immunofluorescence (IIF) [23,31], Western blot (WB) [38,39], and luciferase immunoprecipitation system (LIPS) [25,26]. The following techniques were employed in one paper each: enzyme immunoassay (EIA) [39], particle multianalyte technology (PMAT) [24], and immunoprecipitation [27]. RIPA and LIPS testing methodologies were not based on standardized laboratory techniques. Regarding isotypes, the most frequently investigated was IgG (12/18 papers (67%)), followed by IgA (8/18 (44%)) and IgM (2/18 (11%)). Three papers did not explicitly explore isotypes [25,26,35]. Table 1 summarizes the main characteristics of the studies analyzed in the systematic review.

### 3.2. Sample Collection and Processing

Saliva collection methods were described in most of the studies (17/18 (94%)). In detail, 11 studies described a passive collection of samples through passive drooling [25,26,28,29,30,31,32,33,34,35,39], with eating, drinking, smoking, and tooth brushing restriction described in five papers [25,28,29,30,40]. Tiberti reported specimen collection through prolonged spitting [36]. Palmer and Demoruelle applied a stimulated flow induced by lemon juice or hypertonic saline preparation [27,38]. One study described both passive drooling and stimulated collection through citric acid [25]. Two works described the utilization of a specific device that allows for the collection of the appropriate amount of specimen in two to three minutes [23,24]. Sample purification by centrifugation was described in most of the analyzed studies (11/18 (61%)) [23,24,29,30,31,33,34,35,36,39,40].

### 3.3. Diagnostic Categories

#### 3.3.1. Rheumatologic Disorders

One paper [23] investigated the correlation between saliva and serum antinuclear antibody (ANA) titers for the characterization of a cohort of 70 SLE patients using both indirect immunofluorescence (IIF) and commercial enzyme-linked immunosorbent assay (ELISA) techniques. One previous study conducted by our group [24] evaluated the correlation between saliva and serum testing for the characterization and definition of a cohort of patients affected by heterogenous autoimmune disorders, accounting for a population of 48 subjects. The following analytes were tested: anti-double-stranded DNA antibodies, RNP, Sm, Ro52, Ro60, SS-B, CENP, Scl-70, Ribo-P, Jo-1, and DFS70. Autoantibody specificity and titers were determined with particle-based multianalyte technology (PMAT, INOVA Inc., USA) based on antigen-coated marbles and two classification diodes [24]. Ching [25] and Burbelo [26] analyzed the correspondence between serum and saliva testing results for the identification of patients with sicca syndrome. The former assessed the detection power of classical SjS-related antibodies (anti-Ro52, Ro60, and La) and distinguish 27 affected patients from 27 HC, and the latter explored the sensitivity and specificity of classical antigens (anti-Ro52, Ro60, and La) to identify subjects with sicca syndrome (20 subjects with SjS, 23 subjects with immune checkpoint inhibitor-induced sicca (ICIS), and 20 subjects with autoimmune polyendocrinopathy-candidiasis-ectodermal dystrophy (APECED)) and to determine the detection power of salivary enriched proteins to define APECED subjects with sicca. Both studies applied luciferase immunoprecipitation system (LIPS) technology, which assesses the luciferase activity generated by the bound between antigens and antibodies.

Four papers focused on RA and explored the correlation between serum and salivary antibodies in discriminating patients from controls (including HC and subjects “at-risk” for RA based on familial RA and/or serum anticitrullinated protein antibody (ACPA) positivity) [27,30], the association between antibodies and periodontitis [29], and that between autoantibody specificity and titer and disease activity [28]. In detail, Demoruelle [27] compared serum and saliva results for both anticitrullinated protein antibodies (ACPA) and peptidylarginine deiminase (PAD) enzymes using a commercial ELISA kit and a two-step immunoprecipitation procedure. The aim of the study was to assess the sensitivity of this methodology in distinguishing RA patients from HC and to evaluate of the reliability of precursor markers for RA (anti-PAD-4 antibodies) in subjects at high risk of disease development (defined as subjects “at-risk” of RA based on familial RA and/or serum anticitrullinated protein antibody (ACPA) positivity). Overall, the population was composed of 37 RA patients, 25 HCs, and 46 high-risk individuals. Svärd and Ljungberg [28,29,30] used commercial and modified commercial ELISA kits to determine different ACPA isotypes (IgA, IgA1, IgA2, IgG) and secretory ACPA in both the serum and saliva of 353 RA patients. Due to the utilization of different testing kits, the results were not comparable, and a metanalysis was not possible.

#### 3.3.2. Dermatological Disorders

Two papers focused on PV [31,32,33], accounting for a total of 100 patients without a control group. Both studies applied a commercial ELISA methodology, without the dilution of saliva samples before processing. Results were not comparable due to the application of different positivity cutoffs. Nonetheless, Hallaj [33] demonstrated a stronger correspondence between serum and saliva samples for the detection of both desmoglein 1 (DsG1) and desmoglein 3 (DsG3) compared to Koopaie’s results (Dsg1 serum/saliva, 36/35, Dsg3 serum/saliva 47/47 vs. Dsg1 41/23 and Dsg3 40/25, respectively). The latter applied an IIF methodology in parallel [31]. One study analyzed the correspondence between saliva and serum samples for detection of BP in a cohort of 100 subjects, including 50 BP patients, and 50 HCs [32]. Commercial ELISA testing was applied, and saliva samples were diluted with a 1:2 ratio. The investigated antibodies were bullous pemphigoid antigen II (BP180 NC16a) and bullous pemphigoid hemidesmosomal protein (BP230-C fragment). All included patients were diagnosed after histopathologic evaluation and direct IF study.

#### 3.3.3. Gastrointestinal Disorders

Three studies investigated the relationship between antibody detection and clinical manifestation of PBC [37,39], while one considered the pathophysiological mechanism of the disorder [38]. The manuscripts included a total of 105 patients with a confirmed diagnosis of PBC and 110 controls (both HCs and disease controls). In Lu’s work, antimitochondrial antibody type 2 (AMA-M2) antibodies were assessed through the utilization of a commercial ELISA kit with a dilution protocol for saliva testing [37]. In Ikuno’s paper, the presence of antibodies directed to pyruvate dehydrogenase complex (PCD-E2) was determined through the application of a protocolled immunoblotting test (WB) and enzyme inhibition assay (EIA) in order to quickly screen patients with preclinical PBC status [39]. Palmer assessed the presence of IgA anti-PDC antibodies in the saliva of 55 PBC patients and 28 HCs through the use of a protocolled WB and an in-house ELISA test [38]. One study compared saliva and serum transglutaminase IgA (tTG-IgA) efficacy in distinguishing celiac patients from controls using an ELISA methodology [40].

#### 3.3.4. Endocrinological Disorders

Three papers focused on T1DM detection via saliva antibody assessment, accounting for 101 patients and 180 controls or first disease relatives (34-36). Markopoulos tested a commercial ELISA assay for detection of glutamic acid decarboxylase autoantibodies (GADA) in a cohort of 110 subjects in both saliva and serum samples [35]. In Tiberti’s work, both GADA and antityrosine phosphatase 2 (IA-2) antibodies were assessed in a cohort of 170 individuals using a protocolled radio immunoprecipitation assay (RIPA) [36]. The same technique was applied for GADA detection by Todd in a cohort of 31 T1DM patients. The different arrangements employed in these studies prevented study comparison [34].

In regard to the type of research conducted in eligible studies, most of the papers were cross-sectional (12/18, 66%), three were observational cohort studies (17%), and three were case–control papers (17%).

### 3.4. Saliva Testing as a Diagnostic Tool

Saliva testing showed moderate to good diagnostic accuracy in discriminating patients and controls in most of the studies that compared two such populations (10/12) [23,25,27,28,30,32,35,36,37,38,39,40]. In five studies, it was not possible to assess the ability of saliva testing to distinguish affected subjects from healthy controls due to the lack of adequate controls [24,29,31,33,34]. Furthermore, in one work, saliva testing was assessed only in a subcohort of patients (APECED), not allowing for comparison [26]. In a study that investigated the ability of saliva samples to discriminate patients from controls, Zhang et al. observed that ANA IIF intensity was significantly higher in patients than controls (*p* < 0.01) [23]. A marked difference in antibody titers was also observed in a study by Ching, in which SjS patients and controls were significantly differentiated (*p* < 0.0001); anti-Ro60 and anti-Ro52 salivary antibodies testing presented a high sensitivity and very high specificity (Ro60: 70% sensitivity and 96% specificity; Ro52: 67% sensitivity and 100% specificity) [25].

Similarly, Palmer et al. showed significant diagnostic accuracy in distinguishing between PBC-affected patients and controls in relation to saliva antibody titers of IgA and anti-SC (*p* < 0.0001 and *p* < 0.001, respectively) [38]. Similar results were observed by Ajdani et al. when analyzing the salivary levels of tTG (*p* < 0.001) [40]. Demoruelle et al. and Ljungberg et al. both showed statistically significant differences in terms of antibody titers comparing RA patients and controls (*p* < 0.01; *p* < 0.05) [27,28]. In one study of RA-affected subjects, saliva testing did not prove statistically significant differences in distinguishing patients form HCs and first-degree relatives (*p* > 0.05) [30]. In an effort to distinguish T1DM patients from controls, Markopulos and Tiberti reported the complete or almost complete absence of salivary antibodies in controls and, in contrast, the ability of the detected antibody to correctly identify all affected subjects [35,36]. A similar result emerged in a study by Ikuno in which all controls were found to be negative for PBC-specific antibodies [39]. In a study by Lu, definition of an AUC of 0.88 (95% CI: 0.65–0.93) with a threshold level of 0.61RU/mL allowed for the correct identification of PBC patients with a sensitivity of 81.82% and a specificity of 80% [37]. In a study by Esmaili, the diagnostic power of the tested antibody was not explicated [32].

### 3.5. Serum–Saliva Comparison

In more than half of the selected works, the saliva results were compared with serum testing results (10/18 (55%)) [23,24,25,28,29,31,32,34,36,37]. Five studies evidenced a high correlation coefficient ranging from 0.5 to 0.9 [24,32,34,36,37], while three studies reported a moderate correlation ranging from 0.33 to 0.46 [23,28,31]. In a study by Ljungberg, the correlation varied in relation to the tested antibodies (IgA, R = 0.455; IgA1, R = 0.434; IgA2, R = 0.277; SC ACPA, R = 0.29; IgG ACPA, R = 0.342 (*p* < 0.001)) [28].

Similarly, in a study by Esmaili and Tiberti et al., correlation results varied widely in relation to the investigated antibody type: from R = 0.9 when testing BP180 (serum sensitivity/specificity: 88%/96%; saliva sensitivity/specificity: 87%/96%) to R = 0.5 when evaluating BP230 (serum sensitivity/specificity: 48%/96%; saliva sensitivity/specificity: 77%/62%) (*p* < 0.001) [32]; and from R = 0.749 when testing for GADA (PPV, 91.7%; NPV, 80%) to R = 0.689 when searching for IA-2A (*p* < 0.0001) [36]. Todd at al. showed that the correlation between serum and salivary testing increased from R = 0.67 (*p* < 0.001) to R = 0.85 (*p* < 0.001) when only seropositive GADA subjects were considered [34]. In a study by Koopaie, the saliva–serum correlation ranged between 0.3 and 0.4 in relation not only to the type of detected antibody but also as a consequence of the applied methodology (ELISA: Dsg1 (R = 0.369, *p* = 0.008), Dsg3 (R = 0.463, *p* = 0.001); IIF: R = 0.409, *p* = 0.003) [31].

One study showed a moderate to low correlation between saliva and serum results fort Ro52and Ro60 testing (0.23 < R < 0.3) [25], while another study reported a low but still significant correlation between the two specimens for both IgG and IgA ACPA (R = 0.235, *p* = 0.014; R = 0.208, *p* = 0.030) [29]. In a study by Ikuno, a good to very good correspondence between AMA testing in saliva and serum was observed [39]. In detail, good correspondence was noticed when considering all Ig isotypes (12 positive serum results vs. 9 positive saliva results), while a very good correspondence was detected when solely IgG isotype was analyzed (9 positive serum results vs. 9 positive saliva results). Another paper highlighted a statistically significant association between saliva and serum titers for Dsg1 and Dsg3 antibodies (*p* < 0.001 and *p* = 0.001, respectively) [33]. Dsg1 sensitivity in both serum and saliva samples reached 70%, while that for Dsg3 was higher than 90%.The diagnostic value of salivary results was assessed in a study of immune-mediated thyropathies in which serum tTG was assumed as the gold standard and saliva tTG-IgA detection was found to provide a sensitivity of 80%, a specificity of 98.15%, and an accuracy of 91.67% with an AUC of 0.9309 [40]. Serum and saliva testing were not compared in some mentioned studies [26,27,30,35,38]. The saliva sensitivity and specificity reported in all the analyzed studies are listed in Appendix A.

### 3.6. Association between Saliva Testing and Clinical Manifestations

Salivary ANA testing titers were found to be significantly correlated with disease index and inflammation marker levels in SLE patients [23]. Salivary IgM-ANA levels appeared to be correlated with the PGA (R = 0.24, *p* < 0.05), dsDNA abs (R = 0.35, *p* < 0.01), and SLEDAI (R = 0.3, *p* < 0.05) measures and negatively correlated with serum C3 (R = −0.35, *p* < 0.01) and C4 (R = −0.26, *p* < 0.05); salivary IgG-ANA was correlated with ESR (R = 0.39, *p* < 0.01), SLEDAI (R = 0.25, *p* < 0.05), and dsDNA abs (R = 0.29, *p* < 0.05) and negatively correlated with serum C3 (R = 0.29, *p* < 0.05); and salivary IgA was correlated with dsDNA abs (R = 0.24, *p* < 0.05) and ESR (R = 0.33, *p* < 0.05).

With a focus on the association between salivary testing antibodies and RA manifestations, one study reported an association between anti-PAD4 salivary antibodies and RA disease duration [27], while Ljungberg et al. identified ACPA salivary levels to be statistically associated with disease activity as determined by DAS28 in linear regression analysis (*p*=0.016) [28]. Saliva IgA was more frequently detected in active patients with RA as assessed by higher ESR and DAS28 levels, higher tender joint counts, worse heath assessment questionnaire (HAQ) scores, and patient global assessment (PGA) compared to negative IgA results (*p* = 0.031, *p* = 0.04, *p* = 0.039; *p* = 0.006; *p* = 0.03, respectively).

When considering PV disease area index (PDAI), salivary ELISA anti-Dsg3 demonstrated a moderate association with mucosal PDAI (R = 0.513, *p* < 0.001), and anti-Dsg1 demonstrated a moderate correlation with body–head–neck PDAI (R = 0.477–0.492, *p* < 0.001) [31].

Furthermore, in a study by Hallaji, Dsg-1 antibodies were found to be related to the severity of mucosal lesions (R = 0.496, *p* < 0.001) [33]. When evaluating the total PDAI score, both anti-Dsg1 and anti-Dsg3 were found to have a moderate correlation with the index (R = 0.336, *p* = 0.017 and R = 0.510, *p* < 0.001, respectively) [31]. When evaluating clinical characteristics, anti-Dsg1 antibody titers were found to be higher in the case of the mucocutaneous phenotype (*p* = 0.021) [33].

When analyzing PB and clinical manifestations, a statistically significant correlation was determined between salivary BP230 levels and mucosal involvement (*p* = 0.03) and between BP180 values and the severity of skin disease (R = 0.54, *p* = 0.01) [32]. Considering PBC, in a study by Lu at al., AMA-M2 antibodies were found to be correlated with IL6 and INFγ levels and indirectly correlated with disease activity (*p* < 0.001). In a study by Palmer, antibodies were not found to be correlated with fatigue scale (R = 0.18, *p* = NS) or dry mouth symptoms (R = −0.17, *p* = NS) [38].

Some of the included studies did not investigate the relation between saliva testing and clinical features [24,25,26,29,34,35,36,39,40]. For example, in [29] a clinical manifestation associated with the investigated antibodies, periodontitis, was found to be unrelated to the autoimmune illness that the population in question was suffering from. Table 2 summarizes the ability of saliva testing to discriminate patients from controls, with a comparison of saliva testing with serum testing and clinical manifestation correlations with saliva specimen testing.

## 4. Discussion

Autoimmune disorders represent an extremely heterogeneous group of diseases with similar and frequently overlapping pathogenic mechanisms, such as the presence of disease-specific antibodies, clinical manifestations, and treatment approaches. Therefore, precise disease-specific autoantibody identification represents a crucial step in patient diagnosis, classification, risk assessment, and monitoring.

In the current study, we investigated the feasibility and accuracy of saliva testing as an alternative to the serum detection method. The key driver for this investigation lies in the great potential saliva specimens to guarantee maneuverability and harmless collection. These two features are extremely valuable in patients such as newborns, children, the elderly, and general medically weakened subjects, for whom blood sample collection might present technical difficulties and further debilitate the diseased. Furthermore, utilization of saliva testing as reliable tool for screening and detection of diseases acquires even greater appeal when considering that the diagnostic process of low-prevalence disorders and autoimmune illness still requires the use of referral labs and professionally performed phlebotomy. In our review, we found that the applicability of saliva specimens for patient detection has been tested in various autoimmune disorders with encouraging results; the sensitivity, specificity, and accuracy of saliva were reported to be comparable to those of serum testing in most of the analyzed studies [23,24,25,28,29,31,32,34,36,37].

Most studies proved a significant correspondence between serum and saliva specimens in terms of antibody detection ability [23,24,25,28,29,31,32,34,36,37], although the analysis highlighted heterogeneity among results when considering the rate of agreement [25,28,29,31,32,33,34,36,39], which varied in relation to the type and isotype of the tested antibodies, as well as the applied methodologies. Peptide, protein, steroid, and drug detection and quantification have been proven to be affected by the application of different protocols for sample collection, storage, and processing [41]. No data have been produced to date on the assessment of the presence or absence of discrepancies in antibody detection in the case of application of different types of sample collection (passive drooling, spitting, swab device, etc.) or antibody stability in relation to temperature and storage. Future studies will need to address standardization of methodologies and validation of saliva testing in autoimmune diseases such as SjS in which salivary secretion is impaired. Furthermore, we found an interesting correspondence between the presence of disease-specific autoantibodies and the occurrence of clinical manifestations such as in SLE, RA, PV, PB, and PBC settings [23,28,31,32,33,37]. If confirmed, these data open an additional application for saliva testing, enabling its use for purposes ranging from diagnosis to disease activity monitoring. These observations have the potential to challenge the paradigm, moving from a model of centralized investigations to patient-centered point-of-care testing. A limitation of this study lies in the heterogeneity of saliva collection and testing methods among the selected papers, which did not allow for a pooled analysis. First, only four studies [25,31,33,40] specified the collection timing between 8.00 and 11.00 a.m., while the remaining papers did not mention the time of collection. Although a mild fluctuation of Ig has been described in relation to the circadian clock [42], age, and stress levels [43,44], unclear evidence supports the low sensitivity of antibody determination for samples collected during outpatient department hours, which occur during daylight. Second, the sensitivity and specificity of saliva testing were reported in only five of the analyzed papers [25,32,33,37,40], limiting comparison across studies. When a comparison between saliva and serum testing was possible, as in the studies by Esmaili [32] and Hallaji [33], the results were promising. Despite the limited number of retrieved studies, the available data support the reliability of saliva testing as a diagnostic tool in clinical practice. Nonetheless, more research is needed to confirm these observations. Finally, cutoff values for positivity varied across studies and in relation to the applied methodology, limiting the comparability of the results.

However, saliva can be confirmed as a sensible and comparable sample for antibody testing in autoimmune disorders, as supported by data reported in most of the studies included in this review. Furthermore, as reported in most of the analyzed studies, saliva allows for the distinction of patients from HCs, irrespective of the tested isotype, and the presence of autoantibodies in saliva appears to be directly correlated with clinical manifestations.

## 5. Conclusions

Saliva testing might represent a solid alternative to serum testing for antibody detection in cases of autoimmune disorders. Despite the presence of heterogeneity in relation to the methodologies applied and the types and isotypes of tested antibodies, saliva research should continue, given the promising results reported herein, demonstrate a correlation between detected antibodies and clinical manifestations. While standardization concerns still exist, these observations have the potential to challenge the current paradigm, moving from a model of centralized investigations to patient-centered point-of-care testing.

## Figures and Tables

**Figure 1 ijerph-20-05782-f001:**
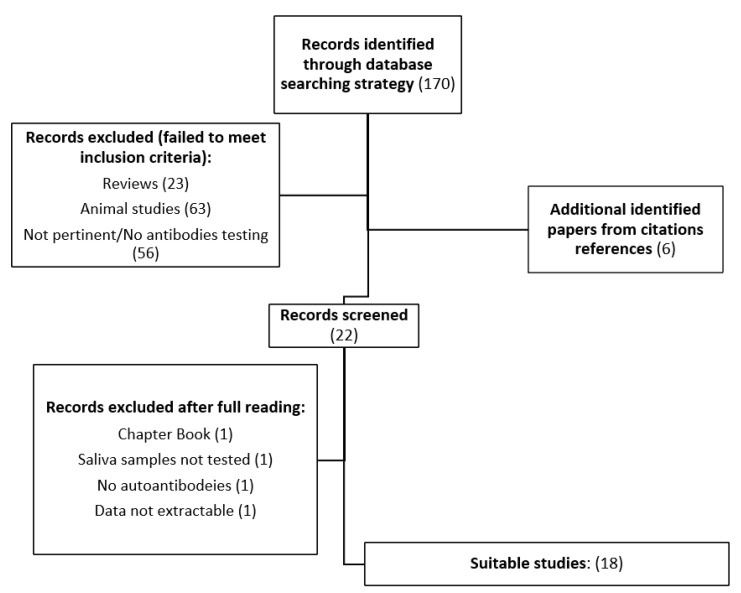
Flow chart of the literature search strategy.

**Table 1 ijerph-20-05782-t001:** Papers included in the final analysis, as well as the tested antibodies and the methodologies applied.

	Author, Year of Publication	Study Design	Population Numerosity	Tested Antibodies and Isotypes	Methodology
CTDs	Zhang, 2021 [23]	C-S	70 SLE, 10 HC	ANA (IgG/M/A)	IIF, commercial ELISA
CTDs	Sciascia, 2022 [24]	O	48 patients (21 SLE, 10 PAPS, 2 SSc, 2 SjS, 8 UCTD, 1 thyroiditis, 4 RA)	dsDNA, RNP, Sm, Ro52, Ro60, SS-B, CENP, Scl-70, Ribo-P, and Jo-1, DFS70 (IgG)	PMAT
Sicca syndrome	Ching, 2011 [25]	C-C	27 SjS, 27 HC	Ro52 and Ro60 (NS)	LIPS
Sicca syndrome	Burbelo, 2019 [26]	C-S	83 (20 HC, 23 ICIS, 20 SjS, 20 APECED)	Ro52, Ro60, La, LPO, BPIFA1, and BPIFA2 (NS)	LIPS
RA	Demoruelle, 2021 [27]	C-S	37 RA, 25 HC, and 46 subjects “at-risk” for RA	anti-PAD4, anti-PAD3/4, and APCA (IgG, IgA)	Immunoprecipitation, commercial ELISA
RA	Ljungberg, 2020 [28]	C-S	196 RA, 101 HC	ACPA (IgG, IgA, IgA1, IgA2, and serum SC)	Modified commercial ELISA
RA	Svärd, 2020 [29]	C-S	132 RA	ACPA (IgA, IgG)	Commercial ELISA
RA	Svärd, 2019 [30]	C-S	25 RA, 21 FDRs, 70 SLE, 11 HC	Secretory ACPA, Salivary IgA ACPA	Modified commercial ELISA
PV	Koopaie, 2018 [31]	C-S	50 PV	Anti-Dsg1-3 (IgG)	IIF, commercial ELISA
PB	Esmaili, 2014 [32]	C-C	50 BP, 50 HC	BP230 and BP180 (IgG)	Commercial ELISA
PV	Hallaj, 2010 [33]	C-S	50 PV	Anti-Dsg1-3 (IgG)	Commercial ELISA
DM	Todd, 2002 [34]	O	31 T1DM	GADA (IgG)	RIPA
DM	Markopoulos, 1997 [35]	C-S	30 T1DM, 80 HC	GADA (NS)	Commercial ELISA
DM	Tiberti, 2009 [36]	O	70 T1DM, 24 FDR, 76 HC	GADA, IA, and 2A (IgG)	RIPA
PBC	Lu, 2017 [37]	C-S	49 PBC, 60 HC, 42 OLP	AMA-M2 (IgG)	Commercial ELISA
PBC	Palmer, 2000 [38]	C-S	44 PBC, 28 HC, 11 PBC patients post liver transplantation	PDC (IgA)	WB, in-house ELISA
PBC	Ikuno, 2001 [39]	C-S	12 PBC, 11 HC	AMA directed versus 2-OAD enzymes (IgG, IgA, and IgM)	WB and EIA
Celiac disease	Ajdani, 2022 [40]	C-C	39 celiac disease, 39 HC	Salivary and serum tTg (IgA)	ELISA (not specified whether in-house or commercial kits were used)

Note: C-S: cross sectional; C-C: case–control; O: observational. APECED: autoimmune polyendocrinopathy-candidiasis-ectodermal dystrophy; BP: bullous pemphigus; FDR: first disease relative; HC: healthy control (includes patients with diseases other than the primarily considered disease); ICIS: immune checkpoint inhibitor-induced sicca; OLP: oral lichen planus; PBC: primary biliary cholangitis; PV: pemphigus vulgaris; RA: rheumatoid arthritis; CTDs: connective tissue disorders; SLE: systemic lupus erythematosus; SjS: Sjogren syndrome; T1DM: type 1 diabetes mellitus. EIA: enzyme inhibition assay; ELISA: enzyme-linked immunosorbent assay; IIF: indirect immunofluorescence; LIPS: luciferase immunoprecipitation system; PMAT: particle multianalyte technology; RIPA: radio immunoprecipitation assay; WB: Western blotting. ANA: antinuclear antibody; dsDNA: double-strand DNA antibody; RNP: antibodies to ribonucleoprotein; Sm: anti-Smith antibody; CENP: anticentromere antibody; Scl-70: antitopoisomerase 1; Ribo-P: antiribosomal P antibody; DFS70: anti-dense fine speckled 70 antibody; LPO: autoantibodies against lactoperoxidase; BPIFA1: BPI fold-containing family A member 1; BPIFA2: BPI fold-containing family A member 2; ACPA: anticitrullinated protein antibody; PAD: peptidylarginine deiminase antibody; Dsg1: desmoglein 1 antibody; Dsg3: desmoglein 3 antibody; BP180: bullous pemphigoid antigen II; BP230: bullous pemphigoid hemidesmosomal protein; GADA: glutamic acid decarboxylase autoantibody; AMA: antimitochondrial antibody; PDC: antipyruvate dehydrogenase complex; 2-OAD: oxoacid dehydrogenase complex; tTG: tissue transglutaminase antibody NS: not specified.

**Table 2 ijerph-20-05782-t002:** Papers included in the final analysis assessing salivary testing strength as disease detection method and correlation with clinical manifestations.

	Saliva Testing Performance *	Saliva–Serum Correlation or Significant Association	Correlation between Saliva Antibody Testing and Clinical Manifestations
Zhang [23]	ANA IF intensities were significantly higher in SLE patients than in healthy controls (*p* < 0.01)	R = 0.33, *p* = 0.0058	IgM-ANA correlated with PGA (R = 0.24, *p* < 0.05), dsDNA (R = 0.35, *p* < 0.01), SLEDAI (R = 0.3, *p* < 0.05) serum C3 (R = −0.35, *p* < 0.01), and C4 (R = −0.26, *p* < 0.05); IgG-ANA correlated with ESR (R = 0.39, *p* < 0.01), SLEDAI (R = 0.25, *p* < 0.05), dsDNA (R = 0.29, *p* < 0.05), and serum C3 (R = 0.29, *p* < 0.05); IgA-ANA correlated with dsDNA (R = 0.24, *p* < 0.05) and ESR (R = 0.33, *p* < 0.05).
Sciascia [24]	N/A	R = 0.73 (95% CI: 0.68–0.76, *p* < 0.0001). The results obtained with saliva were usedto generate an ROC curve using the serum results as a binary classifier: AUC = 0.97	N/A
Ching [25]	A Mann–Whitney U test showed a marked difference in autoantibodytiters between SjS and control groups (*p* < 0.0001). Ro60: 70% sensitivity (95% CI: 50%–86%), 96% specificity (95% CI: 81%–100%); Ro52: 67% sensitivity (CI 46–83%) 100% specificity (CI 87–100%)	Anti-Ro60 and Ro52titers did not correlate quantitatively with serum titers (R = 0.23, R = 0.2).	N/A
Burbelo [26]	N/A	N/A	N/A
Demoruelle [27]	Statistically significant difference in saliva anti-CCP prevalence between RA patients and controls (*p* < 0.01)	N/A	N/A
Ljungberg [28]	Difference between saliva IgA2 levels of RA patients and controls (*p* < 0.05)	Saliva IgA ACPA moderately correlated with serum levels of IgA (r = 0.455) and IgA1 (r = 0.434) and weakly with IgA2 (r = 0.277), SC ACPA (r = 0.29), and IgG ACPA (R = 0.342) (all at *p* < 0.001)	Saliva IgA ACPA-positive samples had significantly higher ESR (*p* = 0.031), DAS28 (*p* = 0.04), tender joint count (TJC) (*p* = 0.039), HAQ (*p* = 0.006), and PGA (*p* = 0.03) than negative samples. In linear regression analysis, salivary ACPA levels were associated with DAS28 (*p* = 0.016). Salivary IgA ACPA was associated with DAS28 after adjusting for disease duration and treatment (*p* = 0.021).
Svärd 2020 [29]	N/A	Low correlation between IgG and IgA ACPA serum/saliva results (R = 0.235, *p* = 0.014; R = 0.208, *p* = 0.030)	N/A
Svärd 2019 [30]	Salivary IgA ACPA was not statistically more prevalent in RA patients compared to HC or FDR (*p* > 0.05)	N/A	N/A
Koopaie [31]	N/A	ELISA: Dsg1 (R = 0.369, *p* = 0.008); Dsg3 (R = 0.463, *p* = 0.001). IIF: (R = 0.409, *p* = 0.003)	ELISA anti-Dsg1-3 showed a moderate correlation with the total PDAI score (R = 0.336, *p* = 0.017; R = 0.510, *p* < 0.001), Dsg3 showed a moderate correlation with mucosal PDAI (R = 0.513, *p* < 0.001), and Dsg1 showed a moderate correlation with body–head–neck PDAI 8 (R = 0.477–0.492, *p* < 0.001).
Esmaili [32]	N/A	BP180(R = 0.9, *p* < 0.001); BP230 (R = 0.5, *p* < 0.001)’ BP180 sensitivity: serum 88%, SALIVA 87%; specificity, 96% vs. 96%. BP230 serum/saliva sensitivity: 48% vs. 77%; specificity, 96% vs. 62%.	A statistically significant correlation was found between saliva BP180 values and severity of skin disease (R = 0.54, *p* = 0.01); BP230 levels correlated with mucosal involvement (*p* = 0.03).
Hallaji [33]	N/A	1, Dsg1 *p* < 0.001, Dsg3 *p* = 0.001Dsg1 sensitivity (serum/saliva): 72/70%; Dsg3 sensitivity (serum/saliva): 94/94%	Anti-Dsg1 titers were significantly higher in the saliva of patients with mucocutaneous phenotype (*p* = 0.021). Salivary anti-Dsg1 antibodies correlated with severity of mucosal lesions (R = 0.496, *p* < 0.001).
Todd [34]	N/A	(R = 0.67, *p* < 0.001; R = 0.85, *p* < 0.001 when only seropositive GADA subjects were considered)	N/A
Markopulos [35]	GADA was found both in the serum and saliva of all diabetic children, allowing the distinction of patients and controls using both serum and saliva testing (*p* < 0.0001)	N/A	N/A
Tiberti [36]	Salivary GADA and IA2 autoantibodies were present in only one control (1.3%, 1/76)	GADA: (R = 0.749, *p* < 0.0001); IA-2A: (R = 0.689, *p* < 0.0001). GADA correctly identified 91.1% of T1DM patients (PPV, 91.7%; NPV, 80%)	N/A
Lu [37]	AUC 0.88 (95% CI: 0.65–0.93). Threshold value of 0.61 RU/mL, 81.82% sensitivity, and 80% specificity	R = 0.63, *p* < 0.001	IL6 and INFγ saliva values were significantly increased in the PBC group (*p*<0.001).
Palmer [38]	A significant difference was detected between the saliva of PBC patients and HC controls, both for IgA and anti-SC titers (*p* < 0.0001; *p* < 0.001)	N/A	No correlation between anti-PDC IgA and severity of dry mouth and fatigue (R = -0.17, *p* = NS; R = 0.18, *p* = NS)
Ikuno [39]	All controls were negative for the tested antibodies	Good correspondence between serum and saliva for all Ig isotypes: 12 (serum) vs. 9 (saliva); perfect correspondence between IgG isotypes: 9 (serum positivities) vs. 9 (saliva positivities)	N/A
Ajdani [40]	Salivary levels of tTG were greater among patients (*p* < 0.001)	compared to serum gold standard. Saliva results showed an AUC of 0.9309, a sensitivity of 98.15%, a specificity of 80%, and a diagnostic accuracy of 91.67%	N/A

* Defined as the ability to significantly distinguish patients from controls based on salivary antibody titer. N/A: not applicable; not mentioned in the paper.

## Data Availability

The data that support the findings of this study are available from the corresponding author, [S.S.], upon reasonable request.

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
