# Peer review of "New Frontiers in Autoimmune Diagnostics: A Systematic Review on Saliva Testing"

_ijerph, 2023, doi:10.3390/ijerph20105782_

Round 1
Reviewer 1 Report
This paper aimed to investigate the current knowledge on autoantibody testing saliva samples in autoimmune disorders. The introduction is very short and should illustrate, at least, the up-to-date literature regarding the potential diagnostic use of saliva whit particular regard to autoimmune disorders. A total of 164 papers (158 are indicated in fig 1) were retrieved for text-mining analysis and only 18 met the inclusion criteria. Results reported the main autoantibodies identified in literature regarding only 4 autoimmune disorders. My concern regards the very restricted area of interest of this article, (only four pathologies were identified and consequently only 26 references are reported). Indeed, I believe this is a short review and not an article. The text mining approach is the same usually used to collect information for each review article and does not make the original contribution requested for a research article.
Author Response
Thank you for your comment. We double checked the number of screened articles and we amended the article as appropriate. Besides, to improve readability, the introduction has been implemented, to read “Saliva testing showed its applicability in different settings, to name: detection of oral cancer such as squamous cell carcinoma [4], identification of infectious diseases (helicobacter pylori[5], hepatitis [6]papilloma virus[7]), hormones monitoring [8–10], screening for chronic kidney disorder [11]. Interesting results are emerging in myocardial infarction [12]and neurodegenerative disorder [13] detection through saliva specimen testing and in diabetes glucose level monitoring [14]. Further, saliva testing increased researches on periodontal disorders [15] and caries prediction[16]. Moreover, saliva testing has been useful for drugs level monitoring, such as in patients with epilepsy [17] and proved its suitability for forensic medicine and toxicology in drugs of abuse testing [18].”
We believe that the purpose of the review is to clarify what are the application and feasibility of saliva testing in autoimmune disorders and that we unravelled existing data in the results presentation and discussion. The goal of this work is to attract researchers’ attention on a very handy and harmless sample, saliva, and on its potential application in different autoimmune disorders. It was out of the scope of this study to systematically review the available evidence on saliva testing across all the medical conditions. Indeed, we aimed to shed some lights in a specific field of potential interest.
Reviewer 2 Report
1. Since the composition of saliva is significantly subject to diurnal fluctuations, it is important to understand whether these factors are taken into account in studies in addition to the method of saliva collection?
2. I would like to learn summary tables for each group of diseases that would summarize information on sensitivity and specificity. If the same metabolite was determined in saliva in different studies, it is interesting to what extent the values of different authors coincide / do not coincide. Nevertheless, the main problem of saliva use lies in the lack of clear protocols for its study and uniform norm/pathology criteria. The value of the review will be significantly increased if specific numerical values (cut-off criteria) for individual diseases are compared.
3. Often, for saliva metabolites, more correlations are observed with clinical manifestations of diseases than with blood parameters in the same patients. Saliva still has an independent meaning and should not be considered only in conjunction with blood. But explanations for the discrepancies in blood composition are also interesting to see. Or although we can assume on the basis of available information.
4. The article is very poorly designed with many typos and extra spaces.
Author Response
REVIEWER 1.
- Since the composition of saliva is significantly subject to diurnal fluctuations, it is important to understand whether these factors are taken into account in studies in addition to the method of saliva collection?
Thank you for the insight. The manuscript has been implemented as follows: “For example, only four studies [25,32,33,39] specified the collection timing between 8.00 and 11.00 a.m., while the remaining papers did not mention it. Even though a mild fluctuation of Ig has been described in relation to circadian clock[42], age and stress levels[43,44], not clear evidence exist to support a degree of heterogeneity in the sensitivity for antibodies determination.”
- I would like to learn summary tables for each group of diseases that would summarize information on sensitivity and specificity. If the same metabolite was determined in saliva in different studies, it is interesting to what extent the values of different authors coincide / do not coincide. Nevertheless, the main problem of saliva use lies in the lack of clear protocols for its study and uniform norm/pathology criteria. The value of the review will be significantly increased if specific numerical values (cut-off criteria) for individual diseases are compared.
Dear reviewer, thank you for you comment. We revised the available data and we implemented the table accordingly. We found a declared sensitivity and specificity only in five papers, to name: Ching, Esmaili, Hallaji; Lu; Ajdani. The reported papers investigated different disorders, indeed a comparison between work was not possible.
Here below we reported a little table with findings from the selected papers, that was not added to the work.
|
|
Authors, year of publication |
Sensibility |
Specificity |
|
CTDs |
Zhang[1] |
UK |
UK |
|
CTDs |
Sciascia[2] |
UK |
UK |
|
Sicca syndrome |
Ching[3] |
Ro60: 70% , Ro52: 67% |
Ro60: 96%; Ro52: 100% |
|
Sicca syndrome |
Burbelo[4] |
UK |
UK |
|
RA |
Demoruelle[5] |
UK |
UK |
|
RA |
Ljungberg[6] |
UK |
UK |
|
RA |
Svärd 2020[7] |
UK |
UK |
|
RA |
Svärd 2019[8] |
UK |
UK |
|
PV |
Koopaie[9] |
UK |
UK |
|
PB |
Esmaili[10] |
BP180: Serum/saliva 88% Vs 87%; BP230 serum/saliva 48% Vs 77% |
BP180 saliva/serum: 96% Vs 96%; BP230: serum/saliva 96% Vs 62%. |
|
PV |
Hallaji[11] |
Dsg1 sensitivity serum/saliva: 72/70%; Dsg3 sensitivity serum/saliva: 94/94% |
UK |
|
DM |
Todd[12] |
UK |
UK |
|
DM |
Markopulos[13] |
UK |
UK |
|
DM |
Tiberti[14] |
UK |
UK |
|
PBC |
Lu[15] |
81.82% |
80% |
|
PBC |
Palmer[16] |
UK |
UK |
|
PBC |
Ikuno[17] |
UK |
UK |
|
Celiac disease |
Ajdani[18] |
98.15% |
80% |
In Esmaili and Hallaji work a comparison in terms of sensitivity and specificity between saliva and serum showed very similar results. These data indicate saliva as a potential valuable tool for antibodies detection. Nonetheless, more literature is needed to grow solid these findings.
Cut-off values for positivity varied across studies and in relation to the methodology applied, limiting the comparability of the results.
- Often, for saliva metabolites, more correlations are observed with clinical manifestations of diseases than with blood parameters in the same patients. Saliva still has an independent meaning and should not be considered only in conjunction with blood. But explanations for the discrepancies in blood composition are also interesting to see. Or although we can assume on the basis of available information.
We agree with the reviewer when stating that the potential value of saliva testing goes beyond a mere surrogate of serum testing. Further studies are needed to explore the diagnostic and, more critically, pathophysiologic role of autoantibodies detected in the saliva across different autoimmune diseases.
- The article is very poorly designed with many typos and extra spaces.
Thank you for your comment. The paper was revised, and typos and incorrect or missing spaces adjusted.
Round 2
Reviewer 1 Report
Dear Authors,
This paper has been presented as a research article, however it does not provide an original work since it is based exclusively on text-mining analysis. This is a serious flaw in my opinion and the minor revisions made by the authors are not sufficient to consider this paper acceptable for publication in IJERPH.
Maybe It could be presented as a review and submitted to another process of reviewing.
Author Response
Comments and Suggestions for Authors
Dear Authors,
This paper has been presented as a research article, however it does not provide an original work since it is based exclusively on text-mining analysis. This is a serious flaw in my opinion and the minor revisions made by the authors are not sufficient to consider this paper acceptable for publication in IJERPH.
Maybe It could be presented as a review and submitted to another process of reviewing.
Dear Reviewer, we modified the publication type from original research article to review.
As any systematic review of the literature, the originality of the works relies in polling together evidence that might be scattered and/or allowing comparison when possible. In this very case, our study provides new insights on the topic of salvia testing in autoimmune diseases by systematically reviewing available evidence and highlighting current gaps. We hope this paper might help to increase researchers’ knowledge and implement stakeholder effort for methodologies standardization.
Reviewer 2 Report
1. Articles included in the final analysis (18 articles) in tables should be sorted either alphabetically or chronologically.
2. Why did the authors not include in the text of the manuscript the table that they cited in response to comments? It is very indicative in terms of the fact that the topic is not developed and there is very little data.
Author Response
Comments and Suggestions for Authors
1.Articles included in the final analysis (18 articles) in tables should be sorted either alphabetically or chronologically.
- Dear Reviewer, we sorted the article by autoimmune disorder, with the hope to increase its readability. We addeda line in table 1 categorizing articles by diseases.
- Why did the authors not include in the text of the manuscript the table that they cited in response to comments? It is very indicative in terms of the fact that the topic is not developed and there is very little data.
- Dear Reviewer, following your suggestion we included the additional table and text (Table S1) in results: “Saliva sensitivity and specificity from all the studies analysed are reported in supplementary Table 1 (S1)”, and discussion, to read:”the sensibility and specificity of saliva testing were reported only in five papers [25,29,32,33,40], limiting the comparison across studies. When a comparison between salvia and serum testing was possible, as in the studies by Esmaili[40] and Hallaji[25], the results were promising. Available data, despite the limited number of number retrieved studies, support the reliability of saliva testing as a diagnostic tool in clinical practice. Nonetheless, more research is needed to confirm these observations. Finally, cut-off values for positivity varied across studies and in relation to the methodology applied, limiting the comparability of the results.”
|
|
Authors, year of publication |
Sensibility |
Specificity |
|
CTDs |
Zhang[1] |
UK |
UK |
|
CTDs |
Sciascia[2] |
UK |
UK |
|
Sicca syndrome |
Ching[3] |
Ro60: 70% , Ro52: 67% |
Ro60: 96%; Ro52: 100% |
|
Sicca syndrome |
Burbelo[4] |
UK |
UK |
|
RA |
Demoruelle[5] |
UK |
UK |
|
RA |
Ljungberg[6] |
UK |
UK |
|
RA |
Svärd 2020[7] |
UK |
UK |
|
RA |
Svärd 2019[8] |
UK |
UK |
|
PV |
Koopaie[9] |
UK |
UK |
|
PB |
Esmaili[10] |
BP180: Serum/saliva 88% Vs 87%; BP230 serum/saliva 48% Vs 77% |
BP180 saliva/serum: 96% Vs 96%; BP230: serum/saliva 96% Vs 62%. |
|
PV |
Hallaji[11] |
Dsg1 sensitivityserum/saliva: 72/70%; Dsg3 sensitivityserum/saliva: 94/94% |
UK |
|
DM |
Todd[12] |
UK |
UK |
|
DM |
Markopulos[13] |
UK |
UK |
|
DM |
Tiberti[14] |
UK |
UK |
|
PBC |
Lu[15] |
81.82% |
80% |
|
PBC |
Palmer[16] |
UK |
UK |
|
PBC |
Ikuno[17] |
UK |
UK |
|
Celiacdisease |
Ajdani[18] |
98.15% |
80% |
Supplementary Table 1 (S1). Sensitivity and specificity from all the selected studies.